# Analysis of Pollution of Phthalates in Pork and Chicken in Taiwan Using Liquid Chromatography–Tandem Mass Spectrometry and Assessment of Health Risk

**DOI:** 10.3390/molecules24213817

**Published:** 2019-10-23

**Authors:** Ming-Yang Tsai, Chang-Hsun Ho, Hong-You Chang, Wei-Cheng Yang, Chuen-Fu Lin, Chien-Teng Lin, Yi-Jing Xue, Jyh-Mirn Lai, Jiann-Hsiung Wang, Geng-Ruei Chang

**Affiliations:** 1Animal Industry Division, Livestock Research Institute, Council of Agriculture, Executive Yuan, 112 Muchang, Xinhua Dist, Tainan 71246, Taiwan; mytsai@mail.tlri.gov.tw; 2Graduate Institute of Bioresources, National Pingtung University of Science and Technology, 1 Shuefu Road, Neipu, Pingtung 91201, Taiwan; 3Department of Anesthesiology, Show Chwan Memorial Hospital, 1 Section, 542 Chung-Shan Road, Changhua 50008, Taiwan; jikeo2003@gmail.com; 4Ph.D. Program of Agriculture Science, National Chiayi University, 300 Syuefu Road, Chiayi 60004, Taiwan; hongyou@ms1.hinet.net; 5Department of Veterinary Medicine, School of Veterinary Medicine, National Taiwan University, 4 Section. 1 Roosevelt Road, Taipei 10617, Taiwan; yangweicheng@ntu.edu.tw; 6Department of Veterinary Medicine, National Chiayi University, 580 Xinmin Road, Chiayi 60054, Taiwan; cflin@mail.ncyu.edu.tw (C.-F.L.); vet540423@gmail.com (C.-T.L.); asdzxc0523@gmail.com (Y.-J.X.); jyhmirn@mail.ncyu.edu.tw (J.-M.L.)

**Keywords:** phthalates, pork, chicken, residues, tolerable daily intake

## Abstract

Phthalates are widely used plasticizers that can cause endocrine disruption, mutagenicity, and carcinogenic effects and can contaminate food through various pathways. Investigations are scanty on phthalate pollution of livestock and poultry meat and their dietary exposure to humans. The present study assessed residual levels of phthalates in unpackaged pork (30 samples) and unpackaged chicken (30 samples) and their relevance to meat consumption and health risks in the Taiwanese population. Phthalate quantity was assessed by liquid chromatography–tandem mass spectrometry; the materials included diisononyl phthalate, diisodecyl phthalate, benzyl butyl phthalate, di-2-ethylhexyl phthalate (DEHP), and di-n-butyl phthalate. The Taiwan Food and Drug Administration (TFDA) has established values of tolerable daily intake (TDI) for the five phthalates. The major compound detected was DEHP, which ranged from 0.62 to 0.80 mg/kg in two pork samples, and 0.42–0.45 mg/kg in three chicken samples. Collectively, 8.33% of the phthalate-residue-containing samples tested positive for DEHP. The concentrations of DEHP were lower than the screening value of 1.0 mg/kg, as defined by the TFDA. Health risk was calculated as the estimated daily intake (DI) for any likely adverse effects; the DI of DEHP residues was <1% of the TDI value. The estimated risk was insignificant and considered to be safe, indicating that there is no risk to the health of Taiwanese population due to meat consumption. However, it is suggested that a phthalate monitoring program in meat should be instituted for any possible effects in future on human health.

## 1. Introduction

Phthalates are common nonreactive plasticizers and are used in many industrial plastics (e.g., polyvinyl chloride and polyvinylidene chloride materials), household products (e.g., paints and glues), personal hygiene products (e.g., cosmetics, lotions, and perfumes), and medical devices to improve the stretchability, workability, and flexibility of such products [1]. The worldwide production of phthalates grew from 6.2 million tons in 2009 to over 8 million tons in 2011 because of their versatility [2]. Phthalates tend to leach readily and are considered common contaminants of the environment and food. The most common phthalates, including benzyl butyl phthalate (BBP), diethyl phthalate (DEP), dimethyl phthalate (DMP), di-2-ethylhexyl phthalate (DEHP), di-n-butyl phthalate (DBP), and di-n-octyl phthalate (DNOP) are widely associated with the production, use, and disposal of food [3]. Thus, human exposure to phthalates occurs predominantly through diet, and contamination may occur when phthalates are released from food production and plastic food packaging. Dietary ingestion, breathing, and dermal absorption are common avenues through which people become exposed to these compounds [4,5]. Phthalates have been prohibited as food additives in the United States, Japan, Taiwan, and the European Union.

Certain phthalate compounds, such as DEHP, BBP, and DBP, are toxic and have estrogenic effects on wildlife, causing damage to reproduction and development. Furthermore, they are suspected to be carcinogenic, teratogenic, mutagenic, and hepatotoxic agents [5,6]. Phthalates have recently attracted attention; they are widely regarded as endocrine-disrupting chemicals that have detrimental effects on reproductive function, neurodevelopment, and thyroid and sex hormone function [1,7]. The most commonly used plasticizer worldwide is DEHP, which is a potential carcinogen in humans, as indicated by animal studies [8,9]. However, no unwanted reproductive or developmental effects were noted with phthalates such as DMP, DEP, and dioctyl phthalate (DOP) [5,10]. As phthalates are toxic to human health, regulations and standards have been established, including a tolerable daily intake (TDI), minimal risk level, and reference dose. Furthermore, the risks related to using a particular plasticizer-contaminated food material can be determined using TDI values, which are helpful for assessing whether the extent of exposure to a given plasticizer is dangerous to human health [3]. The currently used TDI levels in Taiwan, defined in 2011 by the Taiwan Food and Drug Administration (TFDA), by decreasing order of tolerability (mg/kg/day) are 0.5 for BBP, 0.15 for diisodecyl phthalate (DIDP), 0.15 for diisononyl phthalate (DINP), 0.05 for DEHP, and 0.01 for DBP. Exceeding these suggested TDI levels may have toxic effects on human health.

Taiwanese industries use and manufacture a large amount of plastics. The annual production of phthalates is 0.18 million tons, with a yearly demand of 0.1 million tons of phthalates [5]. Ingested phthalates may originate from numerous sources. Few studies in Taiwan have investigated meat as a potential source of phthalates. In May 2011, the TFDA discovered the illegal use of a prohibited industrial plasticizer as a stabilizing emulsifier to reduce costs, instead of more expensive and safer palmitic acid, by certain food-additive distributors. In Taiwan, DEHP has also been used illegally as a clouding agent in products such as jams, jellies, fruit-flavored juices, beverages such as sports drinks and teas, and in nutraceutical pills and powders [3,11]. However, only a few studies have identified the contamination of meats with the same compound. Therefore, this study examined the residues of five phthalates, namely DEHP, BBP, DIDP, DBP, and DINP, in pork and chicken in Taiwan. An estimated daily mean intake of these phthalates was compared with the associated TDI levels defined by the TFDA, which were calculated from contamination levels in pork and chicken samples at a consumption rate and adult body weight relevant to Taiwanese individuals. The estimated daily mean intake was applied to assess the risks of phthalate exposure for Taiwanese consumers.

## 2. Results

### 2.1. Method Validation

Table 1 reveals that the recoveries, repeatability, and limits of quantification (LOQs) of the phthalates spiked in the pork and chicken samples at two fortification levels. The recoveries of BBP, DBP, DEHP, DIDP, and DINP spiking in the pork samples were 98.3%–99.8%, 98.7%–103.2%, 97.8%–101.3%, 99.5%–102.1%, and 98.2%–99.7%, respectively. The triplicate recovery validation of repeatability varied from 1.7% to 8.5% relative standard deviation (RSD). The values of recovery for BBP, DBP, DEHP, DIDP, and DINP spiking in the chicken samples were 97.8%–99.1%, 100.3%–103.4%, 96.2%–99.5%, 101.5%–102.3%, and 97.2%–98.9%, respectively, whereas the RSD of actual recovery varied from 3.1% to 9.3%. Overall, the mean recovery values obtained from fortified samples ranged from 96.2% to 103.2%, with the pork and chicken samples spiking at two concentrations. The RSD determined in a precision study on the fortified samples was <10% for both the pork and chicken samples.

### 2.2. Detection Rates and Phthalate Levels

Overall, 30 pork and 30 chicken samples were analyzed. As listed in Table 2, 0.80 and 0.62 mg/kg of DEHP could be detected in two pork samples. The ratio of positive phthalates in all pork samples was 6.67%. DEHP was detected in three chicken samples at 0.42, 0.43, and 0.45 mg/kg (Table 3). DEHP was detected in 10% of all chicken samples. Overall, the data from 28 pork and 27 chicken samples were not included because no phthalates were detected. The average level of DEHP contamination in the fresh pork and chicken samples was 0.05 and 0.04 mg/kg, respectively. BBP, DBP, DIDP, and diisooctyl phthalate (DIOP) were also not included because they were undetectable in all samples. In total, 8.33% (5/60) of the analyzed samples contained phthalates, testing positive for DEHP.

### 2.3. Estimated Daily Intake of DEHP Compared to TDI

Observed phthalate daily intake (DI) and TDI (based on a body weight of 60 kg) from detected average DEHP levels in meats were assessed to determine risks for Taiwanese adults. The DI levels determined from the average phthalate residue quantities of DEHP in the pork samples were 0.11 and 0.07 μg/kg of body weight/day in Taiwanese men and women, respectively (Table 4). The DI levels of DEHP residues obtained from chicken samples were 0.04 and 0.02 μg/kg of body weight/day in Taiwanese men and women, respectively. Overall, the highest risk of exposure to DEHP was derived from pork consumption, with a TDI level of 0.22% and 0.14% of 0.05 mg/kg of body weight/day for men and women, respectively. The intake of DEHP through chicken was 0.08% and 0.04% of the TDI level for men and women, respectively. The DI values for phthalate residues against the TDI level for Taiwanese men and women were <1%.

## 3. Discussion

To verify whether the validation method recommended by the TFDA for chemically analyzing food samples [12] is appropriate, chemical residues detected in a range of 0.1–1 μg/g must demonstrate an adequate 70%–120% recovery with <15% RSD. Furthermore, the chemical residues measured in the range of 0.01–0.1 μg/g must exhibit an acceptable 70%–120% recovery with <20% RSD. The quantities of spiked analytes (phthalates) used in the current study were 0.01–0.1 ng/g for five phthalates at lower levels and 0.1–1 ng/g at higher spike levels. The recovery values of phthalates were in the range of 96.2%–101.5% (lower spike levels) and 99.5%–103.5% (higher spike levels), with RSDs of ≤8.7% and ≤9.3%, respectively. Concentrations of all analyzed phthalates were in the acceptable range. Studies have reported that phthalate residues observed in food in the range of 10 ng/g–5.5 μg/g have a satisfactory 70%–120% recovery with an RSD of <15% [13,14]. The use of internal standards could enhance recovery by correcting for the loss of analytes during the extraction procedure, particularly for the detection of phthalates in animal-origin matrixes [15,16]. In this study, we followed the analytical method recommended by the TFDA [17], which uses isotopic standards as internal standards. Thus, the proposed method confirmed that the extraction was applicable and sufficiently robust for the analysis of the investigated phthalates in the pork and chicken samples, which met the aforementioned certification guidelines from the TFDA and related studies.

The limit of all analytes in both pork and chicken samples was 40 ng/g. In 2012, the TFDA recommended LOQs to assess contamination of phthalate esters in foods [18]. The LOQ values of BBP, DBP, DEP, DEHP, diisobutyl phthalate (DIBP), DIDP, DINP, DMP, and DNOP in different matrixes were 50 ng/g in milk, milk power, oil-rich foods, and drinks and 100 ng/g in edible oils. However, the certification requirement of the TFDA has not strictly set LOQs for meats to detect the presence of phthalates. The fat content of domestic meat is generally in the range of 2%–20%, including in red meat, poultry, and seafood [19]. In general, the LOQs of the pork and chicken samples used to determine phthalate residues in this study met the requirements for official TFDA certification that LOQs be below 50 ng/g in oil-rich foods. Other methods have been developed for analyzing phthalates in meat by using different apparatuses, such as using LOQs for di-ethylhexyl phthalate (DHEP), DIOP, and di-2-ethylhexyl adipate with gas chromatography flame ionization detection in the range of 30–70 mg/kg [20]. LOQs were used for determining 20 phthalates, such as BBP, DBP, DMP, DEP, DEHP, DNOP, and DINP, through gas chromatography–mass spectrometry at 0.5 mg/kg [21], and LOQs were used for determining DBP and DEHP thorough high-performance liquid chromatography (HPLC) with ultraviolet detection at 0.2 mg/kg [22]. Compared to the LOQs of those studies, in our present analytical method, fewer LOQs were used, and those used were more useful for detecting trace residues of phthalates. Thus, the analytical methods in this study can help to control plasticizer contamination incidents.

Numerous studies on phthalates in foods have been conducted since 2011, when the illegal use of DEHP as a clouding agent in beverages for profit maximization was revealed in Taiwan. Recent studies have reported phthalate contamination in unpackaged meat products (livestock and poultry) in China, with such products testing positive for BBP [23], DBP [24], DINP [25], DBP, dicyclohexyl phthalate (DCHP), DIBP, and DEHP [26]. In a European study, pork and chicken samples were revealed to contain BBP, DBP, DEP, and DEHP in Belgium [27]. In New York, pork and poultry were reported to be contaminated with BBP, DBP, DCHP, DEHP, DEP, DIBP, di-n-hexyl phthalate, DMP, and DNOP [28]. In a study in Cambodia, pork samples tested positive for BBP, DIBP, DCHP, DEHP, DEP, dihexyl phthalate, DIBP, bis (2-methoxyethyl) phthalate, DMP, DNOP, and dipropyl phthalate [29]. In the United Kingdom, DHEP and mono-2-ethylhexyl phthalate were detected in poultry [30]. Yin and Su [31] detected DOP in three samples of packaged pork and chicken in Taiwan. These apparent discrepancies are likely due to dissimilarities among the detection analyses. No official report in Taiwan on residual phthalates in unpackaged pork or chicken has been issued; however, the results of the present study revealed a high detection level of DEHP in fresh meat. Our results resembled those of studies conducted in China [26] and Cambodia [29]. The sample size of the current study was larger and more phthalates were analyzed than in the study by Yin and Su (1996), where only two classes of DBP and DOP were detected. Our study revealed that DEHP is the only phthalate in the pork (Appendix A) and chicken (Appendix A) samples.

DEHP was present at the highest concentration, up to 0.80 mg/kg in pork and 0.45 mg/kg in chicken. Schecter et al. [28] and Cheng et al. [29] have analyzed ≥ 9 categories of phthalates in pork. DEHP was detected at an average concentration of 0.300 mg/kg in these pork samples, with a highest concentration of 0.481 mg/kg. DCHP (0.09 mg/kg) and DEHP (0.09 mg/kg) were detected at their highest concentrations in seven meat products tested in a study that determined the levels of 18 phthalates in poultry [26]. In addition, Bradley et al. [30] and Schecter et al. [28] have analyzed eight and nine phthalates in poultry, respectively. DEHP was detected at a highest concentration of 0.322 mg/kg, (average level: 0.0186 mg/kg) in poultry samples. Compared with these recent reports, the highest detected residual levels of phthalates were in meat samples in our present study; however, our study individually collected and analyzed chicken samples that did not resemble the poultry samples of the cited papers. The TFDA established a cutoff reference level of 1 mg/kg for the swift inspection of contaminated products [3]. Thus, among all samples we analyzed, the DEHP level conformed to the regulations enforced in Taiwan; therefore, such pork and chicken products must be withdrawn from the market because of health concerns from phthalates.

However, our results revealed that exposure to DEHP was considerably higher than exposure to other phthalates among Taiwanese people. Plasticizers present in packaging materials contaminated the packaged food [32]. Samples in the present study were collected without packaging directly from suppliers, thereby eliminating packaging-related contamination. The phthalates detected in the current study may have originated from crops cultivated for feed or may have leached from materials in the production process [33]. This indicated that livestock and poultry feed in Taiwan likely contains DEHP. Future studies are necessary to assess phthalate levels in livestock and poultry feed and to investigate possible sources of contamination.

Several parameter guidelines, such as the TDI and hazard index (HI), enable the assessment of health risks in people [34,35,36]. The HI assesses the health risks of noncarcinogenic harmful effects [34,37]. The TDI has been proposed by the Food and Agriculture Organization, World Health Organization Expert Committee on Food Additives, and the European Commission and was used to determine the phthalate content of the samples in the present study [5,35,38]. As presented in Table 4, these DI values were considerably lower than the TDIs for DEHP recommended by the TFDA [3]. The present results indicated that daily phthalate exposure did not exceed the TDI. As the residual values of phthalates were low, the estimated DI values of DEHP in the Taiwanese population were below the TDI. The TDI values suggested no risk to health from pork and chicken consumption in Taiwan. The estimated risk was nonsignificant considering that the DI was <10% of the TDI [39,40]. Therefore, the low phthalate levels in Taiwanese pork and chicken have no adverse effects on human health. From these findings, we suggest that for Taiwanese consumers, the toxicity of phthalates derived from ingesting farmed pork and chickens is not a risk to human health. Furthermore, the investment of resources to monitor phthalate contamination in pork and chicken is crucial, given the extremely low levels of phthalates detected in the present study.

## 4. Materials and Methods

### 4.1. Chemicals and Reagents

Five phthalate analytical standards (>99%), namely BBP, DBP, DEHP, DIDP, and DINP, were obtained from AccuStandard, Inc. (New Haven, CT, USA). Chromatography-grade n-hexane, acetone, methanol, ethyl acetate, formic acid, and n-pentane were procured from Merck (Darmstadt, Germany). Stable-isotope-labeled internal standards, including BBP-d4, DBP-d4, and DEHP-d4 were obtained from BGR (Pointe-Claire, Quebec, Canada); DIDP-d4 and DINP-d4 from Toronto Research Chemicals Inc. (Toronto, Ontario, Canada) were employed to conduct liquid chromatography–tandem mass spectrometry (LC-MS/MS). To eliminate phthalate contamination from glassware, acetone and n-hexane were used to wash the glassware twice, followed by heating at 200 °C for 2 h in a clean oven [41].

### 4.2. Apparatus and Instrumentation

Samples were prepared by using a centrifuge (Allegra X-22R, Beckman Coulter Inc., Fullerton, CA, USA), a nitrogen evaporator (N-Evap-111, Organomation Associates Inc., Berlin, MA, USA), a vortex mixer (type 37600 mixer, Barnstead Thermolyne LLC, Dubuque, Iowa, USA), and a nitrogen generator (Model 05B, System Instruments Co., Tokyo, Japan). LC-MS/MS consisted of a mass spectrometer (ABI 4000 QTRAP, Applied Biosystems, Foster City, CA, USA) in electrospray ionization (ESI) mode and an Acquity Ultra Performance LC system (Agilent Technologies 1200, Agilent Technologies, Palo Alto, CA, USA). To measure the residual amounts of phthalates in the samples, chromatographic separation was conducted in an analytical Acquity UPLC BEH C18 column (2.1 mm × 100 mm × 1.7 µm, Waters Corp., Milford, MA, USA) equipped with a BEH C18 vanguard guard column (2.1 mm × 5 mm × 1.7 µm,, Waters Corp., Milford, MA, USA). Gel permeation chromatography (GPC) (Waters 515 HPLC pump, 717 autosampler, 2414 refractive index detector, Waters Corp., Milford, MA, USA) was treated further to eliminate potential chromatographic interference of lipids for extraction and cleanup with a clean-up column (400 mm × 25 mm, 200–400 mesh, Bio-Beads S-X3, Bio-Rad Laboratories, Inc., Hercules, CA, USA).

### 4.3. Preparation of Standard Solutions

Stock solutions of individual phthalate standards (100 mg of each analyte) were weighed precisely and dissolved in 100 mL of methanol in volumetric flasks for preparing stock solutions. Preparation of working standard mixtures was done by combining each stock solution type, followed by dilution to 1 mg/L. All solutions were kept at −20 °C and were brought to room temperature prior to use. A series of calibration standards were made with the working standard solutions by serial dilutions to cover a range of 0.1–5 μg/mL.

### 4.4. Preparation of Samples

Taiwanese people prefer purchasing fresh, defatted meat without packaging. Sixty samples (30 pork and 30 chicken) were collected from slaughterhouses and traditional markets in major production areas in Taiwan (including, Taoyuan, Hsinchu, Miaoli, Yunlin, Taichung, Tainan, Changhua, Chiayi, Hualien and Kaohsiung) in 2016 (Appendix A). All samples were stored and carried in glassware, which was one criterion adopted for minimizing potential sources of procedural phthalate contamination. To detect residues of phthalates, modified methods described by TFDA were followed to extract and clean each sample [17]. Briefly, 0.5 g of meat sample was homogenized and transferred into a centrifuge tube followed by the addition of 0.5 mL of n-pentane and 0.1 mL of internal standards (0.1 μg/mL). Extraction of phthalates into a 1:4 (*v*/*v*) mix of n-pentane:methanol (3 mL) was done by shaking at room temperature in the vortex mixer for 5 min, followed by centrifugation of the mixture for 10 min at 3500 rpm. The upper supernatant solution was transferred into another tube and the tissue pellets at the bottom were extracted again with 0.5 mL of n-pentane and 3 mL of a mix of n-pentane:methanol (as described above) and centrifuged. The supernatant obtained was mixed with the first extracted supernatant and dried by evaporation. To the dried residue, 5 mL of n-hexane:ethyl acetate (1:1 *v*/*v*) was added and dissolved, and then transferred to a glass tube, and subsequently cleaned with GPC clean up at 5 mL/min flow rate. Phthalates were fractionated between 8.5 and 15 min and evaporated for dryness by using the nitrogen evaporator. Each resultant dry extract was reconstituted with 1 mL methanol. Finally, all extracts were filtered through a 0.2 µm polyvinylidene fluoride membrane filter (Whatman plc, Maidstone, UK) and placed in an autosampler vial for injection into the chromatographic system.

### 4.5. LC-MS/MS Analysis

The residual levels of the phthalates were determined in a volume of 5 µL injected into the chromatography system. Chromatographic separations were done using gradient elution with eluent-A (0.005 M ammonium acetate) and eluent-B (0.1% formic acid). A mobile phase gradient was developed as follows: Start with 25% B for 2 min (flow rate, 0.30 mL/min), step increase of B eluent to 50% from 2.0 to 4.3 min, linearly increased to 80% B from 4.3 to 9.5 min, increased to 100% B from 9.6 min, and subsequently maintained until 17.0 min.

The mobile phase gradient was changed to 25% B after 17.0 min and maintained until 18 min. The mobile phase flow rate was constant at 0.30 mL/min. Mass spectrometry (MS) was determined in positive ESI modes by monitoring the two most abundant MS/MS (precursor/product) ion transitions by using a multiple reaction monitoring (MRM) mode program for each analyte. MS parameters in the ABI 4000 QTRAP mass spectrometer were as described before [19] and were set as follows: capillary voltage of 3 kV, source temperature of 150 °C, desolvation temperature of 500 °C, cone gas flow at 50 L/h, desolvation flow at 1000 L/h, and collision gas argon pressure at 0.13 mL/min. The identification and quantification mass transitions with optimum collision energies were as follows: *m*/*z* 313→205 (7 eV), 237 (5 eV), and *m*/*z* 313→149 (11 eV) for BBP; *m*/*z* 279→205 (14 eV), and *m*/*z* 279→149 (7 eV) for DBP; *m*/*z* 391→167 (14 eV), 279 (9 eV), and *m*/*z* 391→149 (20 eV) for DEHP; *m*/*z* 447→289 (9 eV), 307 (11 eV), and *m*/*z* 447→149 (25 eV) for DIDP; and *m*/*z* 419→275 (12 eV), 293 (13 eV), and *m*/*z* 419→149 (26 eV) for DINP. The retention times for BBP, DBP, DEHP, DIDP, and DINP were 8.46, 8.24, 11.12, 12.04, and 11.63 min in pork matrixes (Appendix A) and 8.46, 8.30, 11.12, 12.01, and 11.59 min in chicken matrixes (Appendix A). The dwell time for every MRM transition was set at 5 ms. Instrument operation and acquisition of data were done using Analyst 1.6 software (Applied Biosystems).

### 4.6. Quality Assurance and Validation

The described method was validated by estimation of repeatability, recoveries, limits of quantification (LOQs), and range of linearity [42,43]. To establish the repeatability and recoveries, triplicate blank samples were spiked with a standard mixture of the analytes at low and high (50 and 100 ng/g, respectively) concentrations for phthalate analyses. Then the recoveries were calculated in triplicate by comparing the measured concentrations of the spiked samples before extraction with the blanks spiked at the same concentration following extraction. The reproducibility of the procedure was shown as a percentage of the relative standard deviation (RSD%), and the LOQs were calculated as the analyte concentration that generated a peak signal of 3 and 10 times above the background noise in the chromatogram. Evaluation of linearity was done through matrix-matched calibration by employing blank sample extracts and by the addition of the corresponding amount of the working target compound solution at 10–1000 ng/mL concentration. Assessment of linearity of the calibration curves was done by fitting a linear mode, least-squares regression analysis (R2 ≥ 0.990), in the studied concentration range. Samples with concentrations lower than these LOQs were considered undetectable [30,44].

### 4.7. Intake of Phthalates Compared with TDI

For assessing human exposure to phthalate residues in pork and chicken, daily intake (DI) was determined from the mean levels of each measured residual plasticizer and was compared with its TDI value established by the TFDA. The DI was determined by using the equation: DI (mg/kg/day) = (daily consumption [g/day]) × (mean phthalate concentration [mg/g])/(human body weight [kg]) [45,46]. The daily pork and chicken consumption values for Taiwanese men and women (126.2 and 88.2 g; and 52.2 and 31.1 g, for pork and chicken, respectively) were procured from the National Nutrition and Health Survey conducted by the Ministry of Health and Welfare [47]. We assumed an average bodyweight of 60 kg for adults as a dietary reference value [47]. To determine the percentage of TDI relating to meat consumption, the contribution of the DI of phthalates through pork and chicken were estimated as follows: percentage = (DI/TDI) × 100 [45].

## 5. Conclusions

This study reported an efficient and sensitive method for determining phthalates in meat. According to our findings, DEHP was the most abundant phthalate and exhibited low levels under the rapid screening level (1 mg/kg) as per the TFDA in pork and chicken without plastic packaging. The findings reveal that the presence of DEHP in chicken and pork in Taiwan does not amount to severe contamination. Furthermore, the average values of the estimated DI of DEHP were far below the TDI established by the TFDA; the averages ranged from 0.14% to 0.22% for the pork samples and 0.04% to 0.08% for the chicken samples relative to the TDIs for Taiwanese male and female adults, respectively. These results suggest that the phthalate residues detected in this study pose no risk of adverse health effects. We are not aware of any other contaminants in livestock and poultry production that contribute to the daily meat intake and that exceed TDI to a hazardous extent within the general population. Taiwan’s pork and chicken are adequate in quality and are sufficiently safe. Phthalate exposure among adults in Taiwan appears to pose a negligible threat to human health. It is recommended that future studies employ risk assessments of meat consumption for Taiwanese children.

## Figures and Tables

**Table 1 molecules-24-03817-t001:** Recovery, repeatability, and limit of quantification (LOQ) results for spiked phthalate levels in pork and chicken.

Analyte	Spiked Level (ng/g)	Pork			Chicken		
		Recovery (%)	RSD (%)	LOQ (ng/g)	Recovery (%)	RSD (%)	LOQ (ng/g)
**BBP**	50	98.3	2.2	40	97.8	6.7	40
	100	99.8	5.6		99.1	5.2	
**DBP**	50	98.7	1.7	40	100.3	8.7	40
	100	103.2	2.5		103.4	9.3	
**DEHP**	50	97.8	6.1	40	96.2	3.1	40
	100	101.3	8.5		99.5	5.8	
**DIDP**	50	99.5	7.9	40	101.5	6.5	40
	100	102.1	7.5		102.3	8.9	
**DINP**	50	98.2	1.9	40	97.2	6.8	40
	100	99.7	4.8		98.9	7.2	

RSD—relative standard deviation; BBP—benzyl butyl phthalate; DBP—di-n-butyl phthalate; DEHP—di-2-ethylhexyl phthalate; DIDP—diisodecyl phthalate; DINP—diisononyl phthalate.

**Table 2 molecules-24-03817-t002:** Detection ratios of phthalates in pork samples (n = 30) collected during 2016.

Detectable Sample	Targets Detected ^1^	Detectable Ratio (%)	Detected Residues (mg/kg)
1	DEHP	3.33	0.80
1	DEHP	3.33	0.62
Total: 2		6.67	0.05 (Mean) ^2^

^1^ Phthalate compounds detected, including BBP, DBP, DEHP, DIDP, and DINP. ^2^ Estimated from all samples (i.e., samples with detected and undetected phthalate levels).

**Table 3 molecules-24-03817-t003:** Detection ratios of phthalates in chicken samples (n = 30) collected during 2016.

Detectable Sample	Targets Detected ^1^	Detectable Ratio (%)	Detected Residues (mg/kg)
1	DEHP	3.33	0.42
1	DEHP	3.33	0.43
1	DEHP	3.33	0.45
Total: 3		10.0	0.04 (Mean) ^2^

^1^ Phthalate compounds detected, including BBP, DBP, DEHP, DIDP, and DINP. ^2^ Estimated from all samples (i.e., samples with detected and undetected phthalate levels).

**Table 4 molecules-24-03817-t004:** Estimated daily intake (DI) levels of DEHP residues in meat and their percentage of tolerable daily intake (TDI) level in Taiwanese adults.

Meat	DI (μg/kg Body Weight/Day)		DI% of TDI		TDI (TFDA) (mg/kg Body Weight/Day)
	Male	Female	Male	Female	
**Pork**	0.11	0.07	0.22	0.14	0.05
**Chicken**	0.04	0.02	0.08	0.04	0.05

TFDA—Taiwan Food and Drug Administration.

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
