# Peer review of "Analysis of Pollution of Phthalates in Pork and Chicken in Taiwan Using Liquid Chromatography–Tandem Mass Spectrometry and Assessment of Health Risk"

_molecules, 2019, doi:10.3390/molecules24213817_

Round 1

Reviewer 1 Report

Dear Authors, dear Editor,

the draft article “molecules-610096-peer-review-v1” reports on pilot measurements of phthalates in marketed pork and chicken meat in Taiwan. The Authors use LC-MS-MS in a triple quadrupole instrument and deuterium-labelled analogs of four main phthalates as isotope dilution internal standards. The topic of their manuscript matches the interest of the journal section of Molecules. Draft version for review still needs editing, since there are some paragraphs that derive from the template (l. 95-97). In addition, language editing is needed at several points in the text.

One main point in manuscript preparation that deserves attention is the order of sections: the Materials and Methods are at the end. I believe the Authors followed the order in the template. I wonder whether this is an editorial choice of the journal and whether Authors can change it to the much more rational order, which places it immediately after the Introduction. In this case, I would strongly recommend using the traditional development of topics that makes Results statements immediately consequential to the employed methods.

Tables 2 and 3 have no sense: 30 +30 meat samples and four tested phthalates (240 measurements) deserve a better frequency analysis. This is unique data coming from a very representative world area, so most risk assessors will “jump” on your data to make their own data analysis on this important issue. Readers who are not familiar with Taiwan cannot make a picture of the sampling plan, and of how it is representative. I suggest that you disclose actual data in a Supplementary table (by the way, I could not open your Supplementary file). It may be of interest to report on where meat was taken from, and where it came from. Is it representative of urban meat market, or is it a more general issue also for the Country (if it is relevant, you may put in a map with sampling locations). In addition, is it defatted pork or whole pork (chicken has much less fat to store phtalates)? Is it meat ready for cooking or does it still have trimmings?

line 127 2.3. Estimated DI of DEHP compared with TDI this is actually Risk Assessment and should be titled accordingly. As a risk assessor myself, I don’t understand the basis for calculation: how was daily consumption of those meat types estimated? Why only male-female and not also children, a more vulnerable population? Of course, I (and most interested readers) am not aware of food consumption styles in Taiwan, so I expect much more information, if RA is a substantial part of the article. Is it that the Taiwanese eat much less meat than Westerners, and more vegetables, so that their intake is less? If you have the full RA data, it would be very useful that you show it.

As you understand, this valuable work needs to be valorized. I suggest that you can re-assemble the content you already have and make a better point on the more interesting parts of your work. I assume that you have all the information you need to make a much clearer picture of the phthalate issue in Taiwan.

Best regards

Author Response

Comments and Suggestions for Authors:

The draft article “molecules-610096-peer-review-v1” reports on pilot measurements of phthalates in marketed pork and chicken meat in Taiwan. The Authors use LC-MS-MS in a triple quadrupole instrument and deuterium-labelled analogs of four main phthalates as isotope dilution internal standards. The topic of their manuscript matches the interest of the journal section of Molecules. Draft version for review still needs editing, since there are some paragraphs that derive from the template (l. 95-97). In addition, language editing is needed at several points in the text.

Response:

Thank you for this comment. Our manuscript was revised and edited by a native English speaker from a professional academic editing service before the first submission. In addition, the present modified version has also been reedited according to the reviewer’s suggestions. Please refer to our attachment—“English Editing Certificate (ISO 9001:2015 certified).”

Revision:

Additional revisions have been made according to your suggestions (those paragraphs from the revision have been deleted from lines 95–97).

One main point in manuscript preparation that deserves attention is the order of sections: the Materials and Methods are at the end. I believe the Authors followed the order in the template. I wonder whether this is an editorial choice of the journal and whether Authors can change it to the much more rational order, which places it immediately after the Introduction. In this case, I would strongly recommend using the traditional development of topics that makes Results statements immediately consequential to the employed methods.

Response:

The submitted template was provided by the journal Molecules. We believed that the order of the sections was well structured, with first the Introduction, second the Results, third the Discussion, fourth the Materials and Methods, and finally the Conclusions sections. This conforms to the journal requirements.

Revision:

No change has been made to the order of sections in the revision.

Tables 2 and 3 have no sense: 30 +30 meat samples and four tested phthalates (240 measurements) deserve a better frequency analysis. This is unique data coming from a very representative world area, so most risk assessors will “jump” on your data to make their own data analysis on this important issue. Readers who are not familiar with Taiwan cannot make a picture of the sampling plan, and of how it is representative. I suggest that you disclose actual data in a Supplementary table (by the way, I could not open your Supplementary file). It may be of interest to report on where meat was taken from, and where it came from. Is it representative of urban meat market, or is it a more general issue also for the Country (if it is relevant, you may put in a map with sampling locations). In addition, is it defatted pork or whole pork (chicken has much less fat to store phtalates)? Is it meat ready for cooking or does it still have trimmings?

Response:

We gratefully accept your comment and have added two supplementary tables and a supplementary figure to the revision. The present study assessed residual levels of phthalates in unpackaged pork and chicken. Therefore, our meat samples were collected from slaughterhouses and traditional markets in major production areas in Taiwan. The purchase of unpackaged meat reflects the purchasing habits of many Taiwanese people. However, distinguishing between meat samples from urban meat markets and general markets in the countryside is difficult. In addition, the assessments of health risks must consider the dietary habits of Taiwanese people. Thus, these samples were ready to be cooked and were defatted.

Revision:

Following your comment and to avoid descriptions that are too verbose, we have added two supplementary tables and incorporated the related description into the revision (line 259).

Table S1. Detection ratios of phthalates in pork samples (n = 30) collected during 2016.

Sample (no.)

Phthalates

Detectable sample

Collected location

Detectable ratio (%)

Detected residues (mg/kg)

1

-

-

Taoyuan

-

-

2

-

-

Taoyuan

-

-

3

-

-

Taoyuan

-

-

4

-

-

Hsinchu

-

-

5

-

-

Hsinchu

-

-

6

-

-

Miaoli

-

-

7

-

-

Miaoli

-

-

8

-

-

Miaoli

-

-

9

-

-

Taichung

-

-

10

-

-

Taichung

-

-

11

-

-

Taichung

-

-

12

-

-

Changhua

-

-

13

-

-

Changhua

-

-

14

-

-

Changhua

-

-

15

-

-

Yunlin

-

-

16

-

-

Yunlin

-

-

17

-

-

Yunlin

-

-

18

-

-

Chiayi

-

-

19

-

-

Chiayi

-

-

20

-

-

Chiayi

-

-

21

DEHP

1

Tainan

3.33

0.80

22

-

-

Tainan

-

-

23

-

-

Tainan

-

-

24

25

-

-

Tainan

-

-

25

DEHP

1

Kaohsiung

3.33

0.62

26

-

-

Kaohsiung

-

-

27

-

-

Kaohsiung

-

-

28

-

-

Pingtung

-

-

29

-

-

Pingtung

-

-

30

-

-

Pingtung

-

-

Total

2

6.67

0.05 (Mean)

–: Undetectable phthalate compounds, including BBP, DBP, DEHP, DIDP, and DINP.

Table S2. Detection ratios of phthalates in chicken samples (n = 30) collected during 2016.

Sample (no.)

Phthalates

Detectable sample

Collected location

Detectable ratio (%)

Detected residues (mg/kg)

1

-

-

Taoyuan

-

-

2

-

-

Taoyuan

-

-

3

-

-

Taoyuan

-

-

4

DEHP

1

Hsinchu

3.33

0.43

5

-

-

Hsinchu

-

-

6

-

-

Miaoli

-

-

7

DHEP

1

Miaoli

3.33

0.42

8

-

-

Miaoli

-

-

9

-

-

Taichung

-

-

10

-

-

Taichung

-

-

11

-

-

Taichung

-

-

12

-

-

Changhua

-

-

13

-

-

Changhua

-

-

14

-

-

Changhua

-

-

15

-

-

Yunlin

-

-

16

-

-

Yunlin

-

-

17

-

-

Yunlin

-

-

18

-

-

Chiayi

-

-

19

-

-

Chiayi

-

-

20

-

-

Chiayi

-

-

21

DHEP

1

Tainan

3.33

0.45

22

-

-

Tainan

-

-

23

-

-

Tainan

-

-

24

25

-

-

Tainan

-

-

25

-

-

Tainan

-

-

26

-

-

Kaohsiung

-

-

27

-

-

Kaohsiung

-

-

28

-

-

Pingtung

-

-

29

-

-

Pingtung

-

-

30

-

-

Pingtung

-

-

Total

3

10

0.04 (Mean)

–: Undetectable phthalate compounds, including BBP, DBP, DEHP, DIDP, and DINP.      

Figure S1. Sampling locations in Taiwan. Sixty samples (30 of pork and 30 of chicken) were collected from major production areas in Taiwan (including Taoyuan, Hsinchu, Miaoli, Yunlin, Taichung, Tainan, Changhua, Chiayi, Hualien, and Kaohsiung).

4.4. Preparation of samples

Taiwanese people prefer purchasing fresh, defatted meat without packaging. Sixty samples (30 pork and 30 chicken) were collected from slaughterhouses and traditional markets in major production areas in Taiwan (including, Taoyuan, Hsinchu, Miaoli, Yunlin, Taichung, Tainan, Changhua, Chiayi, Hualien and Kaohsiung) in 2016 (Figure S1). All samples were stored and carried in glassware, which was one criterion adopted for minimizing potential sources of procedural phthalate contamination.”

line 127 2.3. Estimated DI of DEHP compared with TDI this is actually Risk Assessment and should be titled accordingly. As a risk assessor myself, I don’t understand the basis for calculation: how was daily consumption of those meat types estimated? Why only male-female and not also children, a more vulnerable population? Of course, I (and most interested readers) am not aware of food consumption styles in Taiwan, so I expect much more information, if RA is a substantial part of the article. Is it that the Taiwanese eat much less meat than Westerners, and more vegetables, so that their intake is less? If you have the full RA data, it would be very useful that you show it.

Response:

The meat consumption data of Taiwanese people in the present study are limited to official information available to date obtained from the National Nutrition and Health Survey (1993–1996) conducted by the Taiwan Ministry of Health and Welfare. A National Nutrition and Health Survey is currently underway (2017–2020) and will be published after 2020 (http://nahsit-form.ibms.sinica.edu.tw/; Accessed October 15, 2019). Thus, no updated data for children could be obtained and applied. Thank you for your suggestion; we will continue to study this correlation in our future work.

Revision:

Following your suggestion, we have added several descriptions in the revision (lines 340–342).

“…We are not aware of any other contaminants in livestock and poultry production that contribute to the daily meat intake, and that exceed TDI to a hazardous extent within the general population. Taiwan’s pork and chicken are adequate in quality and are sufficiently safe. Phthalate exposure among adults in Taiwan appears to pose a negligible threat to human health. It is recommended that future studies employ risk assessments of meat consumption for Taiwanese children.”

As you understand, this valuable work needs to be valorized. I suggest that you can re-assemble the content you already have and make a better point on the more interesting parts of your work. I assume that you have all the information you need to make a much clearer picture of the phthalate issue in Taiwan.

Thank you for your comments and suggestions for the revision. We sincerely considered these comments and have made appropriate revisions to the manuscript.

Reviewer 2 Report

Overall, this manuscript describes a reasonable method development appropriate for quantitating concentrations of a range of phthalates in two key meats, pork and chicken.  The description of the analytical method is adequately detailed and provides a limit of quantitation appropriate for assessing phthalate residues in human foods.  The analyses of 30 samples each of pork and chicken indicate that DEHP was the only quantifiable phthalate, and the detected concentrations of DEHP were well below the Taiwan screening levels for DEHP in foods. Thus, this limited sampling program indicates a low-level of concern that phthalate contamination of pork and chicken is presenting a potential food safety risk.

Specific comments

l.56:  substitute “…are widely associated with the production…” for “adopted”.

l.92-93: The phrase “…stemming from the excessive use of plastic materials” should be deleted in that it isn’t consistent with the actual data, i.e., only DEHP was detected and it was present at concentrations far below the TDI, thus indicating that current uses of “plastic materials” are not presenting a likely risk of excessive food contamination.

l.113-114:  DHEP should be DEHP

l.124:  Header of Table 3 should read “chicken” and not “pork”.

l.187-188:  The statement that the “…present study reveal a high level of DEHP in fresh meat” is incorrect;  the actual data indicate DEHP levels were very low (a small percentage of acceptable levels) and were found in less than 10% of sampled meats.

l.189-190:  The findings of this study do not provide insight in conclusion that “tainted food is the primary source of DEHP intake in Taiwan” and should be deleted or clarified; the present study did not examine the contribution of other sources such as consumer products, cosmetics, etc.

l.192:  DHEP was the “only” phthalate quantifiable in this study, and thus was not the “major” phthalate which infers other phthalates were also present.

l.195:  The Fierens et al study did not “detect” DEHP in meats; rather it only modeled likely concentrations based on the environmental concentrations and associated environment fate characteristics of phthalates.

l.226-230:  The preliminary findings of this study do not suggest, as the last sentence of the abstract states, that there is a pressing need for investment of resources to monitor phthalate contamination of pork or chicken given that the extremely low levels of phthalates detected in this study.

Author Response

Comments and Suggestions for Authors:

Overall, this manuscript describes a reasonable method development appropriate for quantitating concentrations of a range of phthalates in two key meats, pork and chicken. The description of the analytical method is adequately detailed and provides a limit of quantitation appropriate for assessing phthalate residues in human foods. The analyses of 30 samples each of pork and chicken indicate that DEHP was the only quantifiable phthalate, and the detected concentrations of DEHP were well below the Taiwan screening levels for DEHP in foods. Thus, this limited sampling program indicates a low-level of concern that phthalate contamination of pork and chicken is presenting a potential food safety risk.

Response:

Thank you for your affirmation.

Specific comments

l.56: substitute “…are widely associated with the production…” for “adopted”.

Response:

We appreciate the reviewer’s suggestion.

Revision:

We have revised this description.

l.92-93: The phrase “…stemming from the excessive use of plastic materials” should be deleted in that it isn’t consistent with the actual data, i.e., only DEHP was detected and it was present at concentrations far below the TDI, thus indicating that current uses of “plastic materials” are not presenting a likely risk of excessive food contamination.

Response:

Thank you for this comment.

Revision:

To condense the manuscript and shorten the revised version, we have deleted this description.

l.113-114: DHEP should be DEHP

Response:

Thank you for this comment.

Revision:

Additional revisions have been made based on your suggestion (lines 108–110).

l.124: Header of Table 3 should read “chicken” and not “pork”.

Response:

We appreciate this comment.

Revision:

Additional revisions have been made according to your suggestion (Table 3).

l.187-188: The statement that the “…present study reveal a high level of DEHP in fresh meat” is incorrect; the actual data indicate DEHP levels were very low (a small percentage of acceptable levels) and were found in less than 10% of sampled meats.

Response:

We appreciate this suggestion.

Revision:

To add clarity to the manuscript, we have revised these statements (line 182).

“…In the United Kingdom, DHEP and mono-2-ethylhexyl phthalate were detected in poultry [16]. Yin and Su [38] detected DOP in three samples of packaged pork and chicken in Taiwan. These apparent discrepancies are likely due to dissimilarities among the detection analyses. No official report in Taiwan on residual phthalates in unpackaged pork or chicken has been issued; however, the results of the present study revealed a high detection level of DEHP in fresh meat.…”

l.189-190: The findings of this study do not provide insight in conclusion that “tainted food is the primary source of DEHP intake in Taiwan” and should be deleted or clarified; the present study did not examine the contribution of other sources such as consumer products, cosmetics, etc.

Response:

We appreciate this suggestion.

Revision:

Additional revisions have been deleted from this statement.

l.192: DHEP was the “only” phthalate quantifiable in this study, and thus was not the “major” phthalate which infers other phthalates were also present.

Response:

Thank you for this comment.

Revision:

Additional revisions have been made to this statement in the revised manuscript (line 186).

“…These apparent discrepancies are likely due to dissimilarities among the detection analyses. No official report in Taiwan on residual phthalates in unpackaged pork or chicken has been issued; however, the results of the present study revealed a high detection level of DEHP in fresh meat. Our results resembled those of studies conducted in China [34] and Cambodia [37]. The sample size of the current study was larger and more phthalates were analyzed than in the study by Yin and Su (1996), where only two classes of DBP and DOP were detected. Our study revealed that DEHP is the only phthalate in the pork (Table S1) and chicken (Table S2) samples.”

l.195: The Fierens et al study did not “detect” DEHP in meats; rather it only modeled likely concentrations based on the environmental concentrations and associated environment fate characteristics of phthalates.

Response:

We appreciate this suggestion.

Revision:

Additional revisions have been deleted from this sentence.

l.226-230: The preliminary findings of this study do not suggest, as the last sentence of the abstract states, that there is a pressing need for investment of resources to monitor phthalate contamination of pork or chicken given that the extremely low levels of phthalates detected in this study.

Response:

We gratefully accept your comment.

Revision:

We have added and integrated this suggestion to the revision (lines 222–224).

“…The present results indicated that daily phthalate exposure did not exceed the TDI. As the residual values of phthalates were low, the estimated DI values of DEHP in the Taiwanese population were below the TDI. The TDI values suggested no risk to health from pork and chicken consumption in Taiwan. The estimated risk was nonsignificant considering that the DI was <10% of the TDI [46,47]. Therefore, the low phthalate levels in Taiwanese pork and chicken have no adverse effects on human health. From these findings, we suggest that for Taiwanese consumers, the toxicity of phthalates derived from ingesting farmed pork and chickens is not a risk to human health. Furthermore, the investment of resources to monitor phthalate contamination in pork and chicken is crucial, given the extremely low levels of phthalates detected in the present study.”

In addition, our manuscript was revised and edited by a native English speaker from a professional academic editing service before the first submission. The present modified version has also been re-edited according to the reviewer’s suggestions. Please refer to our attachment—“English Editing Certificate (ISO 9001:2015 certified).”

Thank you for your comments and suggestions for the revision. We sincerely considered these comments and have made appropriate revisions to the manuscript.
